# Study on the Effect of Soft–Hard Material Interface Differences on Crack Deflection in Nacre-Inspired Brick-and-Mortar Structures

**DOI:** 10.3390/biomimetics9110685

**Published:** 2024-11-09

**Authors:** Yifan Wang, Xiao Yang, Shichao Niu, Biao Tang, Chun Shao

**Affiliations:** 1School of Mechanical Engineering, Hangzhou Dianzi University, Hangzhou 310018, China; wangyifan415@foxmail.com (Y.W.); shaochun@hdu.edu.cn (C.S.); 2Key Laboratory of Bionic Engineering, Jilin University, Changchun 130022, China; niushichao@jlu.edu.cn; 3Zhejiang Uniview Technologies Co., Ltd., Hangzhou 310051, China; 17366635022@163.com

**Keywords:** nacre, interface, fracture toughness, cohesive

## Abstract

Nacre has excellent balanced strength and toughness. In this paper, the mechanical performance of the typical “brick-and-mortar” structure, including the stress–strain and strain at the interface as well as the stress in the bricks, was calculated by a simplified analytical model of the nacre. This paper proposes a new method to control the crack deflection based on the toughening mechanism of the nacre. The crack extension of the “brick-and-mortar” structure was simulated using cohesive elements based on the traction–separation law with elastic and softening stiffness as variables, and it was found that both stiffness could effectively control the crack extension. The strength and toughness of the models with different stiffness combinations were calculated and plotted as a function of elastic stiffness and softening stiffness, showing that elastic stiffness significantly affects strength and softening stiffness is a determinant of toughness.

## 1. Introduction

Nature usually assembles materials with hierarchical structures to obtain functional properties superior to the inherent properties of their components. The nacreous layer is one such fascinating biological material, comprising brittle inorganic components and softening organic materials, but exhibits almost 3000 times more fracture toughness than its main constituent aragonite [1]. In addition, nacre also has extremely high strength [2]. It is also noteworthy that the structure of nacre is highly complex. The structure of the nacreous layer of the shells of different mollusc species exhibits significant variation [3]. Prior research has demonstrated that the remarkable mechanical properties of the nacreous layer are attributable to its sub-micron brick and mortar structure, which defines it as a natural biocomposite material [4]. Tablet obliquity is very effective in increasing the stiffness of pearl-layered bimodal structures, and also has a relatively high effect on strength and toughness, as well as influencing their energy dissipation mechanisms [5]. The mechanical properties of pearl-like composites characterized by interlocking angles, especially stiffness and strength, show enhancement compared to conventional B-M structures [6]. There are two critical aspect ratios for tablets with a stepped pearl layer structure: a smaller critical aspect ratio tablet undergoes damage fracture and another higher critical aspect ratio keeps the composite strength intact [7]. The most common definition of toughness today is the amount of energy absorbed/dissipated by a material through deformation without destruction. The high toughness comes from the synergetic and harmonious collaboration of various toughening mechanisms [8]. A significant mechanism of toughening is crack deflection in both the meso- and microscale [9]. As selected in most cases in nacre-inspired materials, crack control, by employing weaker interfaces, is considered to be a clever strategy for improving mechanical performance and avoiding catastrophic failure [10]. The role of the organic materials is to provide crack deflection, which makes it difficult for cracks to propagate through the aragonite layer of the nacre [11]. On the other hand, the sliding between tablets increases the viscoplastic energy dissipation of the interfacial biopolymer [12]. The relatively weak interface can be selected with suitable materials, while the energy dissipation can be effectively improved through interface toughening. The biomimetic coating prepared by pulsed laser deposition technology can effectively improve the fracture toughness of the interface [13]. The particle toughener dispersed in the interlayer enrichment region can enhance the delamination resistance [14]. Surface modification methods such as surface pretreatment is also commonly used in interface enhancement [15]. Implementing gradients of interlocking angles in composite design can effectively regulate crack extension paths, which can help stop crack development and delay catastrophic failures, potentially enhancing the mechanical elasticity of composites [6].

Although these studies demonstrate that interfacial toughening does have an effect on the mechanical performance of the material, they do not elucidate the relationship between interfacial toughness and crack deflection. Furthermore, no optimization strategy is provided for interfacial toughening. This paper investigates the influence of interfacial toughness on crack deflection, utilizing the elastic and softening stiffness of the interface as variables. A tensile shear chain model is employed in conjunction with a cohesive finite element model to provide guidance for the design of nacre-inspired composites that satisfy different mechanical requirements.

## 2. Model and Methods

### Biomimetic Model of Nacreous Structure

The fundamental structure of nacre, as illustrated in Figure 1a, comprises specific overlapping regions between adjacent layers, which are characteristic of the basic platelets. During the tensile process, cracks occur and propagate along the soft phase between adjacent platelets, as illustrated in Figure 1b. In light of the aforementioned tensile behavior, a two-dimensional simplified “brick-and-mortar” (B-M) model was constructed for the purpose of investigating fracture performance, as illustrated in Figure 1c [16].

A two-dimensional numerical simulation was conducted on a tensile specimen with dimensions of 150 × 30 μm, wherein each tablet was observed to measure 15 μm in length and 1 μm in thickness. All interfaces are designated as cohesive (viscous) cells with a thickness of 0 μm and a mesh size of 1 μm. The rigid phase cells are represented by bilinear reduced integral planar quadrilateral stress cells (Four-node plane stress element).

The tensile performance of the B-M model can be abstracted as a tensile shear chain (Figure 1d). The hard phase, designated as “Brick,” was considered to be a homogeneous linear elastic material. The softening phase was subdivided into cohesive tensile (CT) and cohesive shear (CS) interfaces, as illustrated in Figure 1d. The failure of the interface was found to follow the traction–separation law (TSL), as specified in the cohesive model [17,18,19]. An illustration of the TSL is shown in Figure 2a, in which the maximum strength of CT and CS are supposed to be the same. TSL includes an elastic increase and a linear softening stage, where damage happens when the stress or strain reaches the defined initial threshold of damage. The cohesive stiffness *K*^ELA^ and *K*^SOFT^ in the elastic and softening stages are calculated by the following equations:(1)$$K_{CT}^{ELA}=\frac{\sigma_{CT}^{lim}}{\delta_{CT}^{lim}},\;K_{CT}^{SOFT}=\frac{\sigma_{CT}^{lim}}{\delta_{CT}^{lim}-\delta_{CT}^{ult}},\;K_{CS}^{ELA}=\frac{\tau_{CS}^{lim}}{\delta_{CS}^{lim}},\;K_{CS}^{SOFT}=\frac{\tau_{CS}^{lim}}{\delta_{CS}^{lim}-\delta_{CS}^{ult}}$$
where $\sigma_{CT}^{lim}$ and $\tau_{CS}^{lim}$ are the interface strength, $\delta_{CT}^{lim}$ and $\delta_{CS}^{lim}$ are the relative displacement at the peak value of traction, and $\delta_{CT}^{ult}$ and $\delta_{CS}^{ult}$ are the critical displacement for complete failure.

The unloading modulus $K^{unload}$ in the softening stage is as follows:(2)$$K^{unload}=DK^{ELA}=\frac{\delta^{ult}-\delta }{\delta^{ult}-\delta^{lim}}K^{ELA}$$
where δ is the relative displacement of an interface before unloading, and damage factor 0 ≤ *D* ≤1.

With regard to the B-M structure, the deformation of the brick is much smaller than that of the cohesive interface, and the deformation of the CT interface, $\delta_{CT}$, is bounded with the adjacent CS interface, which is formulated as follows:(3)$$\delta_{CT}=\delta_{CS}^{left}+\delta_{CS}^{right}$$
where $\delta_{CS}^{left}$ and $\delta_{CS}^{right}$ denote, respectively, the elongation of the CS element on the left and right sides of the CT element.

According to the TSL specified in Figure 2, the cohesive stresses in the CT and CS interface elements are determined by the following equations:(4)$$\sigma_{CT}=\left\{\begin{array}{c} K_{CT}^{ELA}\delta_{CT}(\delta_{CT}\leq \delta_{CT}^{lim})\\ K_{CT}^{unload}\delta_{CT}(\delta_{CT}>\delta_{CT}^{lim}) \end{array}\right.$$
(5)$$\tau_{CT}=\left\{\begin{array}{c} K_{CS}^{ELA}\delta_{CS}(\left|\delta_{CS}\right|\leq \delta_{CS}^{lim})\\ K_{CS}^{unload}\delta_{CS}(\left|\delta_{CS}\right|>\delta_{CS}^{lim}) \end{array}\right.$$

The normal stress of platelets at position *x* can be calculated by the following:(6)$$\sigma_{brick}(x)=\sigma_{CT}+\frac{1}{t}\int_{0}^{x}\tau_{CS}(x)dx$$where *t* denotes the thickness of the platelet. The average tensile stress in the platelet can be calculated by the following:(7)$${\bar{\sigma }}_{brick}(\varepsilon )=E_{brick}{\bar{\varepsilon }}_{brick}=\frac{1}{l}\int_{0}^{l}\sigma_{brick}(\varepsilon ,x)dx$$where ε = ∆/*l* is the average tensile strain of the model, ${\bar{\varepsilon }}_{brick}$ is the average tensile strain in the brick, and $E_{brick}$ is its Young’s modulus.

The Young’s modulus and Poisson’s ratio of the bricks were 100 GPa and 0.33, respectively, and interface properties are exhibited in Table 1 [20]. $G_{Ic}$ and $G_{IIc}$ represent the critical energy release rate (CER) in the normal and shear directions, respectively, and *η* is the cohesion property parameter.

Displacement load was applied to the model along the horizontal direction, and both sides of the model were boundary-coupled to the rigid body to prevent excessive deformation. The fracture toughness of the interface could be modified by adjusting the interface stiffness in the elastic and softening stages, as shown in Figure 2b. The normalized stiffness *K*/*K*^ELA^ and *K*/*K*^SOFT^ for the elastic and softening stages were adopted here as control groups.

## 3. Results and Discussions

The tensile fracture processes of B-M structures with different elastic modulus and softening stiffness are shown in Figure 3. The models exhibit low strength at low elastic stiffness, the strength of the models increases significantly with increasing elastic stiffness, but the increase in strength begins to decrease after twice the normalized elastic stiffness. By increasing the softening stiffness, the strength and toughness of the models can be improved simultaneously. At four times the normalized softening stiffness, there is almost no increase in strength, but the models do not show destructive failure. Thus, it is possible to optimize the softening stiffness in a small range to improve both strength and toughness.

Figure 4 shows the tensile results for interfaces with different fracture toughness obtained by modifying the elastic modulus. These models with low elastic stiffness have a similar damage pattern whose crack expansion path consists of a main crack dominated by shear damage and many microcracks. With an increase in normalized elastic stiffness of up to six times, shear slip and shear microcracking are reduced, leading to tensile damage of platelets pulling out as the main energy dissipation and failure mechanism. A similar damage pattern occurs in the tensile results obtained by adjusting the softening stiffness, and it is worth noting that the fracture toughness is close in both approaches when the damage pattern is significantly changed, which indicates that controlling the fracture toughness can effectively control the crack extension.

The results of the strength and toughness function surfaces with elastic and softening stiffness as variables are shown in Figure 5. From Figure 5a, it can be seen that the strength of the model is closely related to the softening stiffness; if the softening degree is small, the strength of the model increases and the softening stiffness tends to be linear; if the normalized softening stiffness reaches two times when the strength almost reaches the peak of 300 MPa, then the strength is almost no longer high; elastic stiffness also affects the strength to a certain extent, but its influence on the strength is smaller than that of the softening stiffness. Toughness is more dependent on elastic stiffness; the interface elastic stiffness increases up to two times before the toughness almost falls to the minimum value of 1.5 × 10^6^ J/m^3^. After reaching two times, the toughness tends to stabilize. The decline in toughness is mainly due to two reasons: first, the interface elastic stiffness is too high for the strong and weak interface combination to achieve the failure of the crack deflection control strategy; second, it also leads to the weakening of the special pearl layer mechanism, which improves strain toughening through the stress flexure. The effect of softening stiffness on the toughening of the pearl layer bionic structure is smaller than that of elastic stiffness; the softening stiffness mainly improves the toughness by increasing the overall strength of the material, and the strength grows very little after the softening stiffness is doubled, so the toughness is only increased by a small amount with a stable trend.

More data results were obtained by calculating models with more stiffness combinations, and the strength and toughness were set as a function of elastic and softening stiffness. It is explicitly seen that the strength of the model is closely related to softening stiffness, and toughness depends more on elastic stiffness. Therefore, an appropriate reduction in elastic stiffness and an increase in softening stiffness can improve the mechanical performance of nacre-inspired composites.

## 4. Conclusions

An analytical model was used to describe the mechanical properties of a B-M structure simplified by a 3D-stacked nacre structure. Numerical models were also developed in ABAQUS using cohesive elements to capture the crack extension by the traction–separation method. The effect of fracture toughness at different interfaces on the material’s mechanical performance was investigated by adjusting the elastic and softening stiffness. The numerical model shows that adjusting the interface stiffness can control the crack deflection effectively, which also has a significant effect on the strength and toughness of the material. The model presented in this paper is formulated in 2D. As long as the cross-section of the 3D model corresponds to that of the 2D model, the primary conclusions drawn from this study remain valid for the 3D case. Further research is required to validate the applicability of the three-dimensional B-M model. In addition, the overlapping randomness of the tablets in the microstructure and the influence of the interfacial corrugation and roughness on the material can also be discussed by this model, which requires further calculations.

## Figures and Tables

**Figure 1 biomimetics-09-00685-f001:**
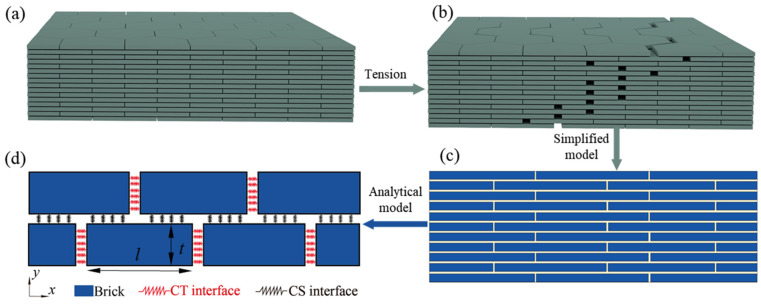
(**a**) Schematic diagram of nacre structure; (**b**) tensile deformation; (**c**) simplified 2D B-M structure; (**d**) analytical model of B-M structure.

**Figure 2 biomimetics-09-00685-f002:**
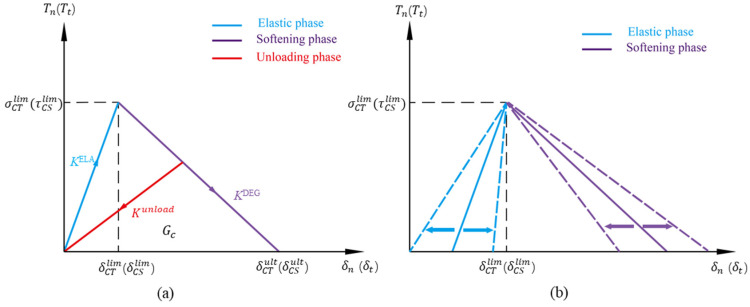
(**a**) Traction–separation law; (**b**) schematic diagram of fracture toughness change.

**Figure 3 biomimetics-09-00685-f003:**
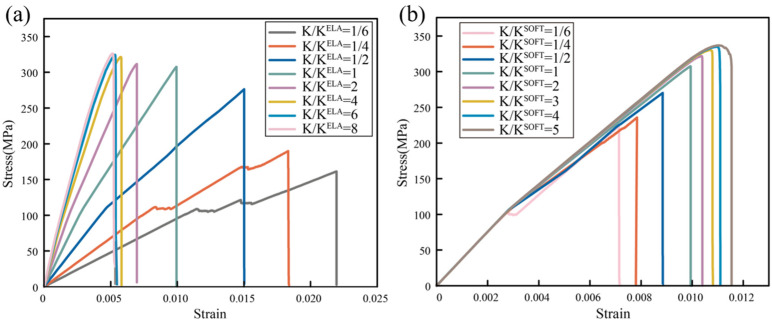
(**a**) The stress–strain curves of the B-M structures with different elastic stiffness. (**b**) The stress–strain curves of the B-M structures with different softening stiffness.

**Figure 4 biomimetics-09-00685-f004:**
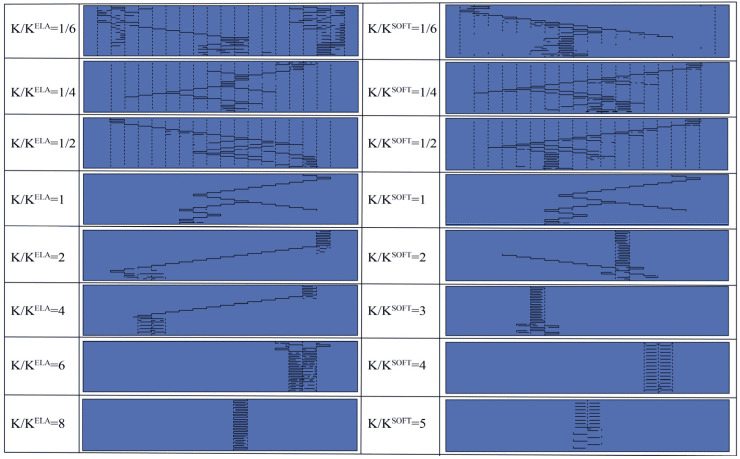
Tensile results with adjusted elastic and softening stiffness. The left side of the figure shows the crack expansion results for various normalized elastic stiffness *K*/*K*^ELA^, and the right side shows the crack extension results for various normalized softening stiffness *K*/*K*^SOFT^.

**Figure 5 biomimetics-09-00685-f005:**
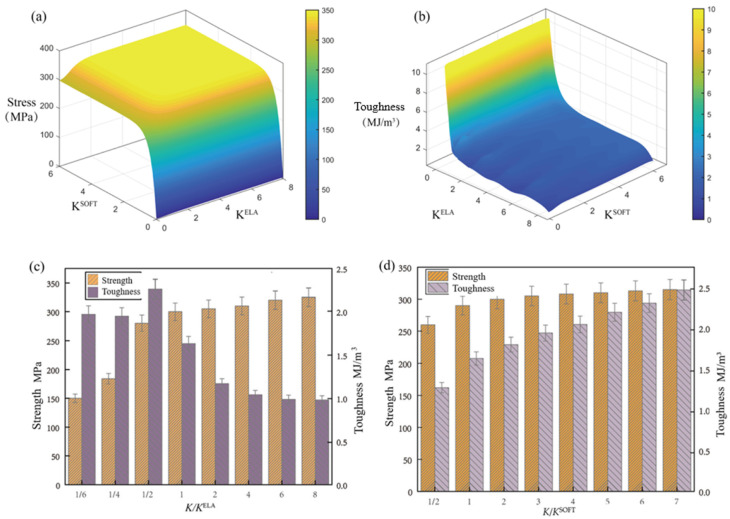
(**a**) Strength as a function of elasticity and softening stiffness. (**b**) Toughness as a function of elasticity and softening stiffness. (**c**) Strength and toughness of different elastic stiffness models. (**d**) Strength and toughness of different softening stiffness models.

**Table 1 biomimetics-09-00685-t001:** Cohesive zone model parameters.

$K_{CT}^{ELA}$ (MPa/mm)	$K_{CS}^{ELA}$ (MPa/mm)	$\sigma^{lim}$ (MPa)	$\tau^{lim}$ (MPa)	$G_{Ic}$ (N/m)	$G_{IIc}$ (N/m)	η
1000	800	30	45	3	5	1.45

## Data Availability

No data were used for the research described in the article.

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
