# Peer review of "Study on the Effect of Soft–Hard Material Interface Differences on Crack Deflection in Nacre-Inspired Brick-and-Mortar Structures"

_biomimetics, 2024, doi:10.3390/biomimetics9110685_

Round 1

Reviewer 1 Report

Comments and Suggestions for Authors

In addition to the issues regarding the references (see above), here are some detailed comments:

This paper addresses an important challenge in the  design of composite materials and structures, that is, how to jointly improve both strength and toughness, two properties that are usually incompatible. The structure described is brick-and-mortar (B-M), inspired by the structure of nacre,  in which highly stiff platelets (Aragonite) are parallel to each other and partially overlapping, bonded at their interfaces by organic soft material. The paper presents analytic and finite element models, both implementing the interfacial material as cohesive zone elements, with different traction-separation laws (TSL) for tension and shear. The models are run for a range of the TSL elastic and softening stiffnesses (rising and declining stress, respectively), demonstrating that simultaneous reduction of the interface elastic stiffness and increase of its softening stiffness can improve both strength and toughness. Crack propagation paths are also shown.

However, the paper is not described in sufficient detail to enable reproduction of the results. To merit publication, the authors need to resolve the following issues (not listed in order of importance):

  1. The paper has many spelling and phrasing mistakes, which require thorough English editing.
  2. The literature survey is quite limited, and ignores classic studies on B&M biological and engineering structures.
  3. The authors do not offer any comparison of their method and results to models published in the literature.
  4. Figure 1 describes the simplified analytical model, but does not describe the essentials of the FEA model (elements size and type, number of elements, boundary conditions, 2D or 3D, meshing, etc.) These should be added.
  5. The definition of the softening stiffness Ksoft (Eq. 1) is such that it is negative. So, what is meant by increasing Ksoft – increasing its absolute value?
  6. Equations 4-6: the conversion from the interface displacement δ to the position x is not clear. My understanding is that the model should compose a system of equations, one for each brick in Figure 1d. Please clarify.
  7. Table 1: please justify the chosen values of stiffness and strength. Are they from the literature? Biological or synthetic materials? Why is the shear strength higher than the tensile strength? Also, provide definition for the symbols G and η.
  8. Specify the source of figures 3, 4 and 5 – which was generated by the analytic model and which by the FEM.
  9. Figure 5: It is suggested to point out on the plots the regions of simultaneous improvement in strength and toughness. Also, provide definition of the toughness.  Also, please add reference to Figure 5 from the text.
  10. The main result - simultaneous improvement in strength and toughness by tuning the interfacial stiffnesses – is not discussed as to the physical mechanisms that govern this tuning. In other words, why elastic stiffness reduction combined with softening stiffness increase are effective in achieving this improvement.
  11. In the conclusions, the authors state that the 2D conclusions apply to 3D as well. Please justify.

Comments on the Quality of English Language

The text should be examined by an English speaking reviewer

Reviewer 2 Report

Comments and Suggestions for Authors

The authors deal with the topic by a numerical point of view only. The study should be implemented with experimental data.

Comments on the Quality of English Language

A global review is recommended

Reviewer 3 Report

Comments and Suggestions for Authors

This research paper consists in a series of calculations the objective of which is to construct a mathematical model expressing the properties of any material mimicking a natural model:  the calcareous structures found in some groups of Mollusks that becomes famous by a specific inter action with light: the Nacre.

The physical reason making nacre interacting with light has been understood only recently: thickness of the nacreous units is in the same magnitude order that visible wave lengths: between 0.4 and 0.7 micrometers. Contrasting to this thickness dimension, the horizontal dimension of the nacreous crystals is ten times superior: between 5 to 10 microns depending on species. Actually, nacreous materials nacre is produced only in the phylum Mollusca, but in different lineages in this phylum. Secretion mode for these nacreous units differs depending on lineages.

Therefore not only nacreous units exhibit different physical properties but from biochemical mode o secretion many specific conditions are still not precisely established.

Therefore the general statement written in the introduction must be modified:   “Nacre has excellent balanced strength and toughness” is not supported by any biological evidence. It relies on vague and imprecise representation of nacre and the rest of the manuscript clearly show the Authors have no knowledge of what a nacreous unit is.

The line 61 statement : ”The hard phase named “Brick” was considered to be a homogeneous linear elastic material” clearly presents the simplifications request for calculation, followed by additional simplification requested for  “CT and CS interface”. But in the resulting calculations the realistic properties of the different nacreous units are not involved.

 Therefore in the chapter 4: Conclusion the first sentence  “An analytical model was used to describe the mechanical properties of a B-M structure simplified by a pearl layer 3d stacked structure” is fully inappropriate. The interesting calculations are applicable to the model structures but not at all to the nacre as biological material

Before such sentence  (line 151) :“Further research is required to validation of the applicability of the three-dimensional nacreous model”  the calculations must be applied to realistic properties of biological nacre, that is much more complex than the  Figure 1 model.

Reviewer recomendation. 

To be published this interesting calculation approach requests suppression of any reference to nacre material as this physical model does not take into account the realistic properties of the biological material.
